# Bioactivity of Cooked Standard and Enriched Whole Eggs from White Leghorn and Rhode Island Red in Exhibiting In-Vitro Antioxidant and ACE-Inhibitory Effects

**DOI:** 10.3390/nu13124232

**Published:** 2021-11-25

**Authors:** Emerson Nolasco, Mike Naldrett, Sophie Alvarez, Philip E. Johnson, Kaustav Majumder

**Affiliations:** 1Department of Food Science and Technology, University of Nebraska-Lincoln, Lincoln, NE 68588-6205, USA; enolasco2@huskers.unl.edu (E.N.); philip.johnson@unl.edu (P.E.J.); 2Proteomics and Metabolomics Facility, Nebraska Center for Biotechnology, University of Nebraska-Lincoln, Lincoln, NE 68588-0665, USA; mnaldrett@unl.edu (M.N.); salvarez@unl.edu (S.A.)

**Keywords:** whole egg, antioxidant, antihypertensive, enriched, peptide, hydrolysate, gastrointestinal digestion

## Abstract

Hen breed, diet enrichment, cooking methods, and gastrointestinal (GI) digestion modulates the bioaccessibility of the bioactive compounds in eggs, but their synergistic role in modulating bioactivity is still unclear. The present study evaluates the effect of hen breed, diet enrichment, and GI digestion on the cooked whole egg-derived peptides in-vitro antioxidant and antihypertensive activities. Standard and enriched whole eggs from White Leghorn (WLH) and Rhode Island Red (RIR) hens were boiled or fried and subjected to GI digestion. Antioxidant activity was measured through oxygen radical absorbance capacity (ORAC) and gastrointestinal epithelial cell-based assays, and the antihypertensive capacity by in-vitro Angiotensin-I Converting Enzyme (ACE) inhibition assay. WLH fried standard egg hydrolysate showed a high ORAC antioxidant activity but failed to show any significant antioxidant effect in the cell-based assay. No significant differences were observed in the antihypertensive activity, although enriched samples tended to have a higher ACE-inhibitory capacity. The peptide profile explained the antioxidant capacities based on antioxidant structural requirements from different peptide fractions, while previously reported antihypertensive peptides were found in all samples. The study validates the importance of physiologically relevant models and requires future studies to confirm mechanisms that yield bioactive compounds in whole egg hydrolysates.

## 1. Introduction

Whole egg protein, lipid, vitamins, and mineral composition enhance its potential as a functional food. Many efforts to enrich the whole egg have observed considerable success in delivering bioactive compounds, such as omega-3 fatty acids, carotenoids, vitamins, and minerals [1,2]. The bioactive compounds have been shown to enhance health and reduce the risk of diseases [3]. Such benefits would, in turn, lessen the burden on national health systems. Studies have shown the effectiveness of plant sterol-enriched functional foods as a strategy for reducing the risk of cardiovascular diseases (CVDs) [4]. An enriched egg is a food product containing nutrients with health-beneficial properties and could also help prevent CVDs [5].

Another approach involves the utilization of food components to enhance their functional role. In this case, egg proteins have been shown to release bioactive peptides after enzymatic hydrolysis [6]. Processes to release bioactive peptides from their parent proteins include microwave, high hydrostatic pressure, high-intensity ultrasound, fermentation, and enzymatic hydrolysis, among others [7,8,9]. Additionally, domestic and physiological processes, such as cooking and gastrointestinal (GI) digestion, naturally hydrolyze proteins into amino acids and peptides [10,11,12]. Moreover, previous studies showed how cooking and simulated GI digestion modulate the egg biological activities [6,13,14,15].

Egg enrichment and hydrolysis have been shown to enhance egg antioxidant activity [14]. Carotenoids improve whole egg antioxidant activity due to the radical or reactive oxygen species quenching capacity through electron transfer, hydrogen abstraction, or addition of a radical species [2,16,17]. The proteolysis of the egg proteins by the digestive enzymes in the human digestive system has also been shown to increase the bioactivities of the egg protein-derived peptides [14,18,19]. The structure of the peptides has proved to be of importance as the amino acid composition influences its potential antioxidant and antihypertensive activities, the latter one primarily related to Angiotensin I-Converting Enzyme (ACE) inhibition from the Renin-Angiotensin-Aldosterone System (RAAS) [20,21,22]. In addition, little is known of how egg enrichment modulates whole egg antihypertensive activity and the production of antihypertensive peptides after GI digestion. However, enriched egg lecithin from egg yolk phospholipids reduces the blood pressure in spontaneously hypertensive rats (SHR) [23]. The results suggested an endothelial nitric oxide synthase (eNOS) regulation and suppression of the mesenteric artery over-reactivity to hormonal stimulation in a hypertension state, a mechanism different from the RAAS pathway [23]. The adequate processing methods which yield a higher content of bioactive peptides after GI digestion could also enable to enhance the health benefits consumers obtain from eggs. 

Nonetheless, little is known of how the hen breed, hen’s diet, and cooking method after GI digestion modulate egg’s biological activity. Understanding the role of these factors is vital in improving the egg’s role as a functional food. Our initial study showed that factors, such as hen breed, hen’s diet, cooking method, and GI digestion, can modulate the bioaccessibility of carotenoids and omega-3 fatty acids after an in vitro GI digestion [10]. However, what is the impact of the bioaccessible compounds on the whole egg bioactivity remains to be elucidated. Therefore, the purpose of this study is to evaluate the effect of hen breed, egg enrichment, and cooking methods on the cooked whole egg hydrolysate antioxidant activity and the peptide’s antihypertensive effect through in vitro methods.

## 2. Materials and Methods

### 2.1. Materials

Randomly Methylated Beta Cyclodextrin (RMCD) was purchased from Cyclodex (CTD, Inc., Alachua, FL). Trolox (238813), Fluorescein (F6377), 2′,7′-Dichlorodihydrofluorescein diacetate (DCFH-DA)(D6883), 2,2′-Azobis(2-methylpropionamidine) dihydrochloride (AAPH)(440914) was purchased from Sigma Aldrich (St. Louis, MO, USA). The ACE Kit-WST (A502-10) was purchased from Dojindo Molecular Technologies, Inc. (Kumamoto, Japan). Gastrointestinal epithelial cells (Caco-2: ATCC^®^ HTB-37™) and Eagle’s Minimum Essential Medium (EMEM) (ATCC^®^ 30-2003™ were purchased from ATCC (Manassas, VA, USA). Penicillin-Streptomycin (15140122), Hanks’ Balanced Salt Solution (HBSS)(14025-076), and fetal bovine serum (FBS)(10437028) were purchased from Gibco (Waltham, MA, USA). Black and clear bottom 96 well plates (165305) were purchased from ThermoFisher Scientific (Waltham, MA, USA). TPP T75 flasks (TP90026) were purchased from Midsci™ (Valley Park, MO, USA). If not stated, chemicals were reagent grade and purchased from Fisher Scientific (Fair Lawn, NJ, USA).

### 2.2. Whole Egg Preparation and Simulated Gastrointestinal Digestion

Fresh White Leghorn (WLH) and Rhode Island Red (RIR) eggs from standard and enriched diets were collected from the Poultry barn at the University of Nebraska-Lincoln and stored as previously described [10]. The hens diet consisted of a corn-soybean-based diet without (standard) and with flaxseed and ORO GLO^®^ as a source of omega-3, lutein, and zeaxanthin (enriched). The hens were fed weekly for 16 weeks. Next, whole eggs were boiled or fried and subjected to simulated GI digestion following the methods of Minekus et al. (2014) and Mat et al. (2016), with modifications according to our previous studies [10,24,25]. Eggs were boiled in a saucepan with water covering up to 2.5 cm above the eggshell. After 10 min, the eggs were removed and cooled off in running tap water for five minutes. The egg was peeled for storage at −20 °C under vacuum in a vacuum bag. Fried eggs were previously homogenized and cooked in a pan fryer heated at 191 °C (350 °F) for 90 s on each side. The eggs were cooled for 15 min and stored at −20 °C under vacuum in a vacuum bag. The simulated digestion consisted of an oral phase using α-amylase and simulated salivary fluids at pH 7 for 2 min, a gastric phase using pepsin and simulated gastric fluid at pH 3 for 2 h, and an intestinal digestion using pancreatin and simulated intestinal fluid at pH 7 for 2 h. After simulated GI digestion, the whole egg hydrolysate was frozen at −80 °C before freeze-drying. The samples were handled and processed under dim light to minimize any photooxidation of egg compounds. The dried whole egg hydrolysate was stored at −20 °C, protected from light and used for further assays.

### 2.3. Measurement of In Vitro Antioxidant Activity through Oxygen Radical Absorbance Capacity Method

The oxygen radical absorbance capacity (ORAC) method measured the whole egg hydrolysate antioxidant activity. The hydrolysate hydrophilic and hydrophobic extract preparation and their antioxidant assay followed Remanan and Wu (2014) and Nimalaratne et al. (2011) [12,14]. A 50 mL Erlenmeyer flask contained 50 mg of whole egg hydrolysate was used for the analysis, 10 mL of hexane-dichloromethane (1:1) was added to each flask and shaken in an orbital shaker for 1 h at 25 °C, followed by centrifugation at 1693× *g* for 5 min at 4 °C (Thermo Fisher Sorvall Legend X1R, Waltham, MA, USA). The supernatant was collected and evaporated under nitrogen flush in amber color vials, obtaining the lipophilic extract. The lipophilic extract was dissolved using 250 µL of acetone and 750 µL of 7% RMCD in 50% acetone-water solution (1:1; *v*/*v*). RMCD acted as a solubility enhancer for the lipid fraction. RMCD was also used to dissolve the Trolox standards and the blank of the hydrophobic fraction assay.

Consequently, the pellet was dried with a nitrogen flush, dissolved in 10 mL 80% ethanol, vortexed, and transferred to a 50 mL Erlenmeyer flask. The hydrophilic extraction process repeated the shaking and centrifugation steps of the hydrophobic extraction process while using the supernatant for the assay. The results for the hydrophilic and hydrophobic assays were combined for the total ORAC antioxidant capacity. The Trolox Equivalent Antioxidant Capacity (TEAC) units are mM, and the results reporter at TEAC (mM)/g of hydrolysate. All samples were extracted in duplicate and assayed in triplicate.

### 2.4. Measurement of In Vitro Antioxidant Activity in Gastrointestinal Epithelial Cells

The antioxidant activity of the egg white-derived samples was performed according to the cellular antioxidant activity (CAA) assay [26,27]. For this assay, the whole hydrolysate was ultrafiltrated (Catalog # UFC900324, Millipore Sigma, Burlington, MA) to obtain the <3000 Da peptide fraction, the possible bioavailable peptides. Gastrointestinal epithelial cells (Caco-2: ATCC^®^ HTB-37™) were grown in T75 flasks using EMEM supplemented with 1% Penicillin-Streptomycin (*v*/*v*) cocktail and 20% FBS (*v*/*v*) at 37 °C and 5% CO_2_ in a humidified condition. Caco-2 cells were seeded in black and clear bottom 96-well plates at a density of 20,000 cells/well. After reaching 95% confluency, the cells were washed twice with HBSS previously tempered at 37 °C and synchronized using EMEM supplemented with 1% Penicillin-Streptomycin cocktail (*v*/*v*) and 1% fetal bovine serum (*v*/*v*) for 24 h at 37 °C. After synchronization, the cells were washed once with HBSS and incubated with 100 µL of 50 µM DCFH-DA and 100 µL of ≤3000 Da peptide fraction at 25, 50, 100, 250, and 500 µg/mL concentration based on previous optimization (Appendix A). The DCFH-DA and peptides were incubated for 1 h at 37 °C and 5% CO_2_ in a humidified condition. After incubation, the cells were washed twice with tempered HBSS and replaced with 50 µL of tempered HBSS. AAPH was used as an oxidizer with a concentration of 600 µM per well. The AAPH solution was added with an automatic dispenser of the Synergy H1 microplate reader (Biotek, Winooski, VT, USA) by dispensing 50 µL of the AAPH solution for a total volume of 100 µL per well. The fluorescence was measured using an excitation and emission wavelength of 485 and 528 nm, respectively, at 1-min intervals for 60 min. The data points were adjusted to the blank, which did not have inhibitor or AAPH with 25 µM of DCFH-DA. The area under the curve (AUC) was calculated for the inducer control (no inhibitor), inhibitor control (10 µM quercetin), and the peptide treatments. The cellular antioxidant activity was calculated according to Equation (1):

Equation (1) Cellular antioxidant activity formula.
(1)$$\mathrm{CAA}\;\mathrm{unit}\;\left(\%\right)=\left(100-\frac{\int \mathrm{SA}-\int \mathrm{BA}}{\int \mathrm{CA}-\int \mathrm{BA}}\right)\times 100$$

∫SA is the integrated area under the curve for the sample fluorescence or positive control (10 µM quercetin), and ∫BA and ∫CA are the integrated areas under the curve from the blank and the negative control [28]. Two independent samples were analyzed in duplicate, considering an average of three technical replicates per sample per assay. 

### 2.5. Measurement of ACE Inhibitory Activity

The ACE Kit-WST measured the whole egg hydrolysate ACE inhibitory activity [29]. A whole egg hydrolysate solution was prepared by preparing a 1000 µg/mL solution and vortexed for 1 min. The solution was centrifuged at 4600× *g* for 10 min at 4 °C and filtered with a 0.45 µm nylon syringe filter. A 6-point inhibition curve was prepared, and each dilution was mixed with the ACE substrate 3HB-GGG (3-hydroxybutyrate glycylglycylglycine) and the ACE enzyme in a 96-well plate, followed by incubation at 37 °C for 1 h. In this step, the substrate was cleaved into 3HB-G and G-G and then into 3HB and G. Afterwards, an indicator working solution was added to each well and incubated at room temperature for 10 min. The indicator solution used 3HB to reduce the tetrazolium salt, 4-[3-(4-iodophenyl)-2-(4-nitrophenyl)-2H-5-tetrazolio]-1,3-benzene disulfonate sodium salt (WST-1) through a chain reaction into formazan. The ACE-inhibition was measured indirectly by the formazan concentration at 450 nm [29,30]. The regression analysis determined the ACE inhibition IC_50_ value for each sample in µg of whole hydrolysate/mL. The assay was performed in duplicate with two independent assays per sample. 

### 2.6. Identification of the Peptide Profile through UPLC-MS/MS

The whole egg hydrolysate <3000 Da fraction was collected through ultracentrifugation (Catalog # UFC900324, Millipore Sigma, Burlington, MA, USA) and freeze-dried for further analysis. This work was done by the Proteomics and Metabolomics Facility at the Nebraska Center for Biotechnology at UNL. An aliquot of the dried samples was resuspended in water at a concentration of 20 µg/µL. For the hydrophilic interaction liquid chromatography (HILIC) separation of the peptides with 2–3 amino acids, the samples were further diluted 10 times for a 10-µg injection. The separation of the peptides was done on a BEH-Amide 1.7 µm (2.1 × 100 mm, Waters) using a Vanquish (Thermo) high-performance liquid chromatography (HPLC) at 40 °C. A flow rate of 300 µL/min was used with a gradient of A (0.1% formic acid in 100% LC-MS grade water) and B (0.1% formic acid in 100% acetonitrile), as follows: 90% B for 2 min, 90% to 40% B in 12 min, back to 90% in 1 min. The data was acquired on a Q Exactive™ HF mass spectrometer (MS) (Thermo Fisher, Waltham, MA, USA) operating in data-dependent acquisition (DDA) mode, using a mass range of 60 to 750 *m*/*z* on single charged ions. The isolated ions were further fragmented using collision induced dissociation (CID) of an isolation window of 2 m/z. The acquired data were analyzed using PEAKS studio (Bioinformatics Solutions Inc., Waterloo, ON, Canada) to perform de novo sequencing of the chromatograms, integrate the peaks for quantification and normalize based on the total ion current (TIC). 

Reverse-phase chromatography (RPC) was performed for the separation of the larger peptides, samples were diluted 200 times and run using an online peptide separation by first desalting peptides on a trapping column (C18 Pepmap100 0.3 × 5 mm, 5 µm, 100 Å) at 5 µL/min in 1% acetonitrile, 0.1% formic acid before separation into the mass spectrometer using a 75 µm × 25 cm peptide CSH C18 130 Å, 1.7 µm nano-column (Waters) using a linear gradient run at 260 nL/min from 5% B to 32% B over 36 min. Solvents: A is 0.1% formic acid in LC-MS grade water, and B is 0.1% formic acid in 80% acetonitrile. The Q Exactive™ HF MS was run in a DDA mode triggering on peptides with charge states 1 to 4 over the mass range of 375–1500 m/z. All MS/MS (MS2) samples were analyzed using PEAKS studio (Bioinformatics Solutions Inc., Waterloo, Canada). PEAKS studio was set up to search the UniProt Gallus gallus UP000000539 database (20190617, 27,804 entries), assuming no specific digestion enzyme. Peptides were searched with a fragment ion mass tolerance of 0.020 Da and a parent ion error tolerance of 5.0 ppm. The data was normalized based on the TIC. No amino acid modifications and enzyme specificity were considered in the analysis due to the sample preparation process.

The data were exported to Excel and the peptides sequences were subjected to structural requirement constraints of antioxidant peptides. The constraints were the presence of Tyr/Trp or Leu/Phe/Ile in the amino-terminal with the presence of Lys, Leu, His, Trp, Met, Tyr, and Glu in any position at the carboxylic end [9,31,32,33]. Both conditions need to be met to be counted as a potential peptide with antioxidant activity. The intensity of the peptides that complied with the structural requirement constraints was added for a total intensity sum per treatment. Similarly, the peptides sequences were subjected to structural requirement constraints of ACE-inhibitory di-, tri-, and oligopeptides, as described by Wu et al. (2006) [20,34]. The peptide needed to comply with all the constraints to be counted as a potential ACE-inhibitory peptide and a total intensity sum was determined as well. As a quality cutoff, an ALC of 50% for the de novo analysis and a −10logP of 15 for UniProt Gallus gallus UP000000539 database (20190617, 27,804 entries) search was used in the data analysis. The HILIC and RPC data were analyzed separately. The data obtained is descriptive as no replicates were performed for the different treatments analyzed. 

### 2.7. Statistical Analysis

A three-way ANOVA was performed to evaluate significant differences and interaction of hen breed, diet enrichment, and cooking methods Section 2.3 and Section 2.5 using Tukey’s post-hoc test at a *p* < 0.05 significance level. A one-way ANOVA was performed to evaluate significant differences in Section 2.4 using Tukey’s post-hoc test at a *p* < 0.05 significance level. The data are expressed as mean ± standard deviation (SD). The number of replicates is described in the figure captions.

## 3. Results and Discussion

### 3.1. Antioxidant Capacity of Whole Egg Hydrolysates through Oxygen Radical Absorbance Capacity (ORAC)

No difference in the ORAC antioxidant capacity was observed between WLH and RIR counterparts, as shown in Figure 1. A hen breed × cooking interaction (*p* = 0.0009) was observed in the three-way ANOVA. As an interaction result, higher antioxidant activity was observed for RIR boiled standard and enriched samples compared to RIR fried enriched, while no difference was observed for WLH standard and enriched samples in both cooking methods. Finally, the highest antioxidant capacities observed were WLH standard fried (371 TEAC/g of hydrolysate) and RIR standard boiled (360 TEAC/g of hydrolysate) eggs, with no significant difference from their enriched counterparts.

### 3.2. Measurement of In Vitro Antioxidant Activity in Gastrointestinal Epithelial Cells

Previous studies have confirmed the different antioxidant capacity of bioactive compounds, such as peptides from chemical assays compared to in vitro cell models [35,36]. The CAA model is based on the quenching of peroxyl radicals generated by AAPH [27]. For this purpose, we obtained the <3000 Da peptide fraction from the whole hydrolysate as small molecular weight (MW) peptides are absorbed and transported by Caco-2 cells monolayer or ex vivo in Wistar rats intestinal sacs [37,38,39]. Furthermore, previous studies have shown a higher in vitro antioxidant capacity for peptides with a MW between 200 to 3000 Da from chickpea protein hydrolysate [40]. For this assay, we selected the samples with the highest antioxidant activity in ORAC, which were WLH fried standard and RIR boiled standard. The concentrations also consider a daily dietary consumption of approximately two whole eggs for the 500 µg/mL concentration based on 50 µg of the <3000 Da fraction in a well area of 0.32 cm^2^ for a total of 156.25 µg/cm^2^. Accounting for the small intestine area of ~27 m^2^, it would require approximately 43 g of cooked egg to reach such concentration [41]. Our results showed that no significant ROS peroxyl radicals were inhibited at 25, 50, 100, 250, and 500 µg/mL concentrations compared to the inducer control (AAPH) for WLH fried standard and RIR boiled standard <3000 Da peptide fraction in Caco-2 cells. Furthermore, the CAA was significantly lower for all the tested concentrations when compared to the inhibitor control (10 µM Quercetin). The CAA for 100 and 500 µg/mL was 7% and 23% for WLH fried standard and 18% and 24% for RIR boiled standard, respectively.(Figure 2) Our CAA results for the 500 µg/mL concentration are in accordance with previous studies which reported a CAA of 22–35% for 100 µg/mL and 25–33% for 500 µg/mL of <10 kDa peptide fractions from common bean milk and yogurt in Caco-2 cells [26].

### 3.3. ACE Inhibition Capacity of Whole Egg Hydrolysates

A significant difference in whole egg hydrolysate ACE inhibition capacity was found between RIR fried standard among all the treatments, as shown in Figure 3. A significant hen breed × cooking (*p* = 0001) and diet × cooking interaction (*p* = 0.0113) was observed through a three-way ANOVA. The sample with the highest ACE inhibitory activity (lowest IC_50_) value was observed in RIR boiled enriched egg (IC_50_: 221 µg of whole hydrolysate/mL) (Figure 3). The RIR boiled enriched egg high ACE-inhibitory activity and antioxidant activity compared to RIR fried enriched egg suggest a possible synergistic activity of the hen breed and cooking method. Regarding WLH, fried enriched eggs showed the highest ACE inhibition activity value (IC_50_: 272 µg of whole hydrolysate/mL) (Figure 3). These samples are expected to have shorter peptides, a characteristic often found in ACE-inhibitory peptides [31]. For example, white shell standard fried eggs subjected to GI digestion in a 5% (w/v) slurry heated at 80 °C showed the presence of tri- and pentapeptides with antihypertensive activity in fried whole egg digest [6]. The IC_50_ in our study was higher than the observed 50 and 20 µg of total dry matter/mL for the boiled and fried whole egg pepsin-pancreatin digest, respectively. Possible reasons for the difference are the sample preparation or the assay sensitivity to determine the ACE-inhibition of a hydrolysate [6]. In a follow up study, the standard fried whole egg hydrolysate was able to reduce the blood pressure when fed to SHR but not the non-hydrolyzed fried egg [13]. Nitric oxide-dependent vasorelaxation was restored and plasma angiotensin-II (Ang-II) levels were decreased, the latter one suggesting a role in the RAAS pathway [13].

### 3.4. Whole Egg Hydrolysate LC-MS/MS Peptide Profile and Peptide’s Structure-Function Relationship

Samples WLH fried and RIR fried standard along with WLH boiled enriched had a higher number of peptides with 5 AA residues, as shown in Figure 4A. In addition, the distribution of the peptide intensity sum is higher in the five AA residues for each sample, as shown in Figure 4B. We must clarify that the peptide intensity is a relative measure of peptides abundance since it is not a direct quantification of the peptides, and it is dependent on the peptide’s amino acids sequence. The sample with the highest intensity of peptides in the range of 4 to 11 amino acid residues was WLH fried standard. The intensity sum for peptides was mainly driven by a group of 10 or more peptides in all the samples at an intensity level of 10^8^ and two or less at 10^9^ in RPC and HILIC. An exception was noticed for WLH fried enriched were the tetrapeptides LLDR, LDLR, QLGL, VCPF, and AMPF for de novo and 17 pentapeptides had an intensity level of 10^9^. Several studies have reported the antioxidant activity of peptides once released from ovalbumin, such as AEERYP, DEDTQAMP, DSTRTQ, DKLPG, DVYSF, and ESKPV [42,43]. We assume that WLH fried standard high ORAC antioxidant activity, as observed in Figure 1, is linked to the peptide intensity and structure. In contrast, the samples with low ACE-inhibitory IC_50_ values did not show a higher number of di- and tripeptides, contrary to our expectations. Therefore, the structure-function of the peptides could have a major relevance than the peptide intensity in the antihypertensive capacity of the whole egg, as previously described [20].

Figure 5A describes the number of hits and intensity of the peptides, which complied with the antioxidant activity structural requirements. The samples with high intensity in Figure 5B had a higher number of peptides that complied with the antioxidant activity structural requirement. WLH fried standard showed a higher intensity of pentapeptides with antioxidant activity structure, possibly due to the WLH fried standard peptides’ high intensity in Figure 4B, suggesting a relationship between intensity and structure-function with the high ORAC value. However, the peptide profile of RIR boiled standard and enriched samples, which showed a high antioxidant capacity through ORAC, did not show a high intensity of peptides with antioxidant structural constraints. It is proposed that the ORAC antioxidant activity observed in RIR boiled standard and enriched hydrolysates is due to the synergistic effects of peptides, carotenoids, or both in the hydrolysate, although further research is required to determine such an effect. 

The structural requirement constraints for ACE-inhibitory peptides were the following: dipeptides required amino acid residues with bulky and hydrophobic side chains as Phe, Tyr, or Trp in both positions, tripeptides favored aromatic amino acids for the carboxyl terminus, positively charged amino acids in the middle residue, and hydrophobic amino acids for the amino terminus [20]. Tetrapeptides favored Tyr, Phe, or Cys in the carboxyl terminus, Phe in the second residue from the carboxyl terminus, Arg, His, Trp, or Phe in the third residue, and Val, Ile, or Trp in the fourth residue [34]. Similarly, oligopeptides favored Tyr or Cys in the carboxyl terminus, His, Trp, or Met in the second residue from the carboxyl terminus, Ile, Leu, Val, or Met in the third residue, and Trp in the fourth residue [34]. It was observed that only dipeptides, except for VRFP, complied with all the constraints from the accounted literature. However, di- and tripeptides with previously reported ACE-inhibitory activity were found in the samples, as shown in Table 1. The MS2 fragmentation profile of the matched peptides is included in Appendix A. A minimum ALC of 50% and a parent ion mass accuracy of less than ±5 ppm was obtained for all the peptides.

## 4. Discussion

Our previous study reported how egg’s bioactive compounds as omega-3 fatty acids, xanthophylls, and peptides bioaccessibility are influenced by hen breed, hen’s diet, cooking method, and simulated gastrointestinal digestion [10]. The study showed no difference in the peptide content and linolenic (C18:3) fatty acid between the treatments after digestion among enriched samples. However, RIR enriched boiled and fried digested samples had a significantly higher docosahexaenoic (C22:6) fatty acid content compared to WLH. The lutein content after digestion was significantly higher than its standard counterpart only for RIR fried digest [10]. The synergistic effect of these compounds and their interaction, if any, is not well understood yet. The present study exhibits that the in vitro antioxidant capacity assessed through the ORAC method showed no difference between WLH standard and enriched counterparts in both cooking methods. A similar study found no significant difference between white shell boiled, fried, and microwaved egg yolks antioxidant activity through ORAC when fed a wheat-based and corn-based diet [12]. However, the antioxidant activity did decrease when compared to its raw counterpart, which was not evaluated in our study [12]. The hen breed × cooking interaction observed in the reduced ORAC antioxidant capacity in RIR enriched fried egg is proposed to occur due to the high heat treatment during frying, responsible for degradation, isomerization, or oxidation of antioxidants, such as carotenoids [16]. Additionally, it has been reported that heat treatment reduces aromatic amino acids from egg yolk, responsible for its high antioxidant activity [12]. Other studies have observed an increase in the antioxidant activity of whole egg digest when cooked through boiling and frying methods whether digested as a 5% slurry or as a whole egg [14,43]. The increase in antioxidant activity has been attributed to the release of antioxidant peptides and amino acids [9,12,43]. Additional experiments are needed to fully understand if any interaction between peptides and dietary compounds as carotenoids play a role in the whole egg antioxidant activity. Initial studies addressing this challenge found that the type and concentration of phytochemicals and peptides influence the antioxidant activity in ORAC and an antioxidant cell-based model [50,51]. While the peptide structure-function relationship was not clear, the study suggests that the phytochemical structure could influence the interaction modulating the antioxidant activity [51]. Furthermore, the interaction of peptides with phenolic compounds seemed beneficial while assessing the cellular antioxidant activity in Caco-2 cells [50].

Cell models offer insight into the bioactivities of food compounds often not detected through chemical methods [36]. The cellular antioxidant activity assay represents a more physiologically relevant model to measure antioxidant activity from peptides that are bound to the cell membrane or absorbed by the cells [26]. The difference between chemical and cellular antioxidant activity can be observed in the CAA assay, where the samples with the highest antioxidant activity in the ORAC assay were unable to reduce the ROS production compared to quercetin, a radical quencher phytochemical [27]. Rao et al., in 2020, suggested that high molecular weight peptides from egg white can potentially exhibit a higher antioxidant activity in ORAC due to their radical quenching capacity independently of their bioavailability [35]. In a physiologically relevant model, the peptides need to be absorbed by the cell to confer their antioxidant activity and quench the intracellular ROS measured in the CAA assay [27]. Further studies are required to determine the number of peptides that can be absorbed by Caco-2 cells and if these possess any antioxidant activity. 

The difference of our white shell standard fried egg hydrolysate antihypertensive capacity from Majumder and Wu (2009) is possibly due to the hydrolysis of the fried egg digestion in a 5% slurry and 80 °C heat treatment, which could improve the release of antihypertensive peptides [13]. Our hypothesis is also supported by the study of Jahandideh et al. (2014), in which the non-hydrolyzed fried egg was not able to reduce the blood pressure in SHRs compared to the fried egg hydrolysate. In our study, the INFOGEST gastrointestinal protocol follows a more physiologically relevant model of human digestion [52]. Low molecular weight peptides generated by enzymatic hydrolysis are key bioactive compounds that inhibit the ACE enzyme. Rapeseed-derived di, tri, and tetrapeptides have been shown to reduce important RAAS components, such as ACE, renin, Ang-II, and Ang-(1-7), in myocardial tissues [53]. Nonetheless, in our study, no <3000 Da fractionation was performed in the ACE inhibition assay, and it remains to be confirmed if the low molecular peptides are responsible for the ACE-inhibitory capacity. Future experiments are required to confirm if fried whole egg hydrolysate can modulate different components of the RAAS pathway besides ACE to understand the potential of the egg as a functional food. The mechanism by which these peptides are obtained in the enriched samples remains to be elucidated. It has been shown that carotenoids can form a carotenoid-protein complex with xanthophylls, such as astaxanthin and omega-3, can interact with proteins when forming gels [54,55]. Furthermore, carotenoids as lutein in salads have been observed to undergo micellarization during simulated digestion with no influence from triglycerides added to the salad [56]. It is proposed that the carotenoids interact to a higher degree with the egg lipid fraction, leaving the protein fraction accessible for hydrolysis. However, further studies are required to confirm this hypothesis and if omega-3 fatty acids have any effect on the release of antihypertensive peptides.

The peptide size distribution showed no difference of small MW peptides number and intensity through de novo analysis complying with the antioxidant structural requirements (Figure 4A,B). Suggesting that the previously observed ORAC antioxidant capacity of WLH fried standard could be due to peptides with a MW of >3000 Da with the structural constraints described in Section 2.6. WLH fried standard was not able to reduce the AUC in the cellular antioxidant activity assay, and its CAA was significantly lower than 10 µM, supporting the difference previously reported between chemical and in vitro cell model antioxidant assays [36]. WLH fried standard high MW peptides could be more efficient in scavenging peroxyl radicals through hydrogen atom transfer reactions in vitro in ORAC since it is independent of physiological processes, such as absorption, metabolism, and antioxidant responsive signaling pathways, to name a few [36]. 

It is assumed, based on our results that peptides having at least two of the mentioned structural characteristics could potentially have ACE-inhibitory activity. Previously reported peptides generated through pepsin-pancreatin enzyme combination, such as Asn-Phe (NF) and Tyr-Arg (YR) from ovalbumin, and Met-Pro-Phe (MPF) from ovotransferrin, were found in most of the samples [6,47,48]. The peptides were generated through the pepsin-pancreatin enzyme combination as previously reported. Peptides from different enzymes or combinations, such as Leu-Trp (LW) (pepsin), Leu-Tyr (LY) (chymotrypsin), and Ala-Trp (AW) (chymotrypsin-thermolysin), were also found. It remains to be elucidated if the ACE-inhibitory capacity is derived from the individual peptides or by a synergistic effect of the peptide’s mixtures along with other molecules present in the whole hydrolysate. Furthermore, a recent study using a quantitative-structure relationship (QSAR) model built with an artificial neural network showed that the C-terminal is of primary importance for ACE-inhibitory dipeptides when containing a hydrophobic amino acid [57]. As shown in Table 1, the dipeptides which did not comply with both structural requirements did have a hydrophobic and aromatic amino acid in the C-terminal, supporting our assumption. 

Finally, hen breed and cooking interaction factors play a role in the in-vitro ORAC antioxidant activity where RIR fried enriched antioxidant activity was lower than its boiled counterpart and fried standard egg. Similarly, ACE-inhibition with the whole hydrolysate tended to be better in RIR boiled samples compared to boiled, although no significant difference was observed and the tendency did not hold for WLH samples. Following previous studies, we found a discrepancy between chemical and cell-based antioxidant capacity [35,36]. Caution should be taken when interpreting the physiological relevance of whole egg hydrolysates chemical antioxidant capacity, which depends on inherent pathways and metabolism of the cells in question. Characterization of the peptide profile supported the results by showing a higher number and intensity of pentapeptides in WLH fried standard samples, as well as peptides complying with the antioxidant structural constraints. However, the ACE-inhibitory capacity was independent of the intensity and number of peptides in the hydrolysate, suggesting the peptide structure-function relationship plays a bigger role. Further studies are needed to confirm the role of lutein and omega-3 in the release of antihypertensive and antioxidant peptides. Research on the stability and absorption of whole egg hydrolysate <3000 Da peptides into gastrointestinal epithelial cells to implement their bioactivity is of utmost importance to understand the difference between chemical and cell-based antioxidant capacity. The novel knowledge of the study highlights how factors’ interactions, such as hen breed and cooking, can modulate the whole egg bioactivity in vitro, but further studies are required to confirm the mechanisms to release bioactive peptides and their physiological relevance on in vivo studies. 

## Figures and Tables

**Figure 1 nutrients-13-04232-f001:**
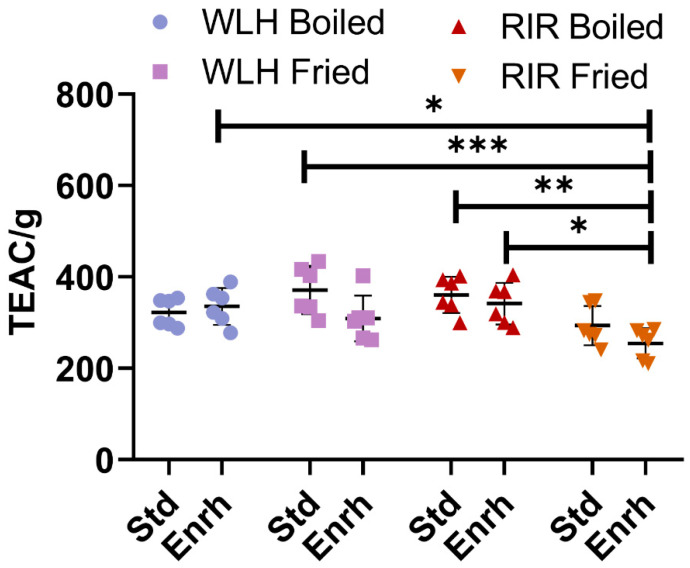
Total ORAC antioxidant capacity of whole egg hydrophilic and hydrophobic extract. Error bars represent standard deviation (SD); *, **, and *** Statistically significant difference (*p* < 0.05, *p* < 0.01, and *p* < 0.0001, respectively, three-way ANOVA, Tukey’s post-hoc test), *n* = 6. Note: Trolox Equivalent Antioxidant Capacity (TEAC/g of hydrolysate), Std: Standard, Enrh: Enriched.

**Figure 2 nutrients-13-04232-f002:**
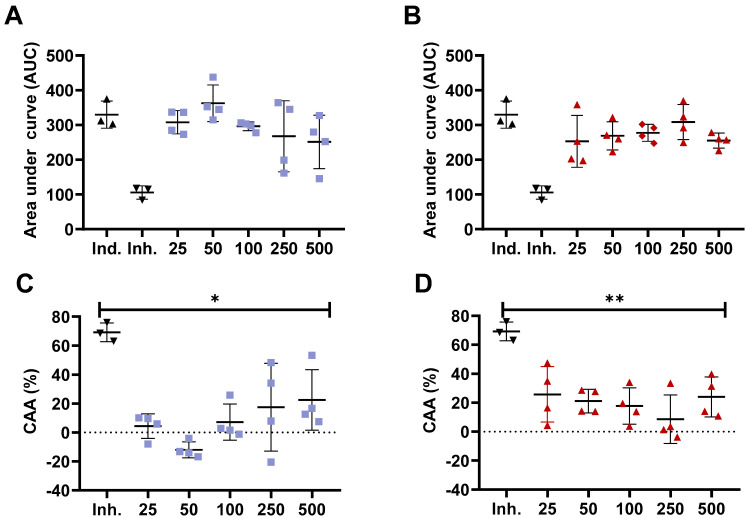
Cellular antioxidant activity (CAA) of White Leghorn Fried standard and Rhode Island Red boiled standard <3000 Da peptide fraction in Caco-2 cells. Area under the curve (**A**,**B**) and cellular antioxidant activity (**C**,**D**) for White Leghorn (**A**,**C**) and Rhode Island red (**B**,**D**). Concentrations are expressed in µg/mL. Ind.: Inducer control (600 µM AAPH + 25 µM DCFH-DA), Inh.: Inhibitor control (10 µM quercetin + 600 µM AAPH + 25 µM DCFH-DA). * and ** Statistically significant difference (*p* < 0.05, and *p* < 0.01, one-way ANOVA, Tukey’s post-hoc test), *n* = 4. Error bars represent standard deviation (SD).

**Figure 3 nutrients-13-04232-f003:**
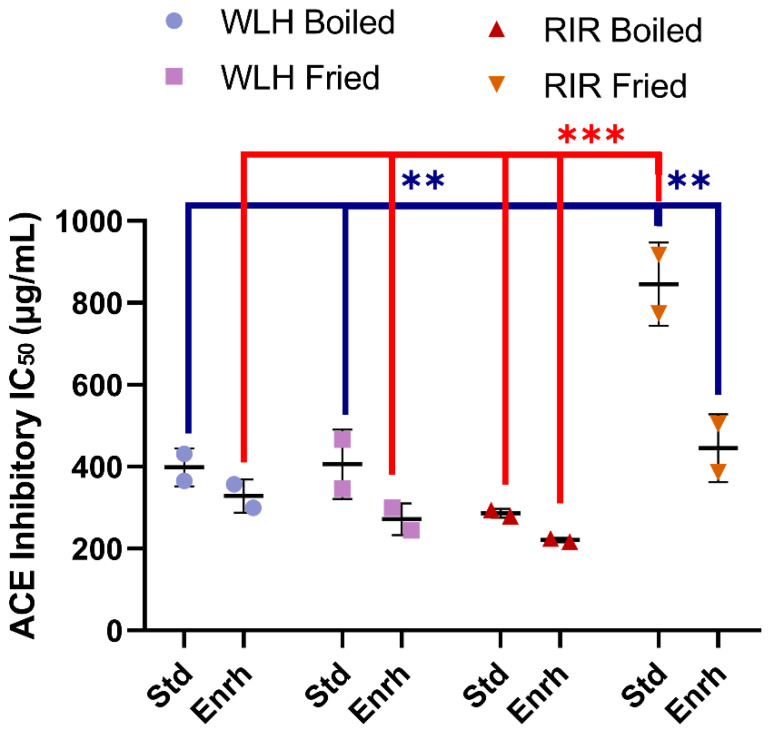
Whole egg hydrolysates ACE Inhibition IC_50_ (µg of whole hydrolysate/mL). Error bars represent standard error of the mean (SEM); *, **, and *** Statistically significant difference (*p* < 0.05, *p* < 0.01, and *p* < 0.0001, respectively three-way ANOVA, Tukey’s post-hoc test), *n* = 2. Error bars represent standard deviation (SD). Note: IC_50_: Protein concentration which inhibits 50% of ACE activity, WLH: White Leghorn, RIR: Rhode Island Red, Std: Standard, Enrh: Enriched.

**Figure 4 nutrients-13-04232-f004:**
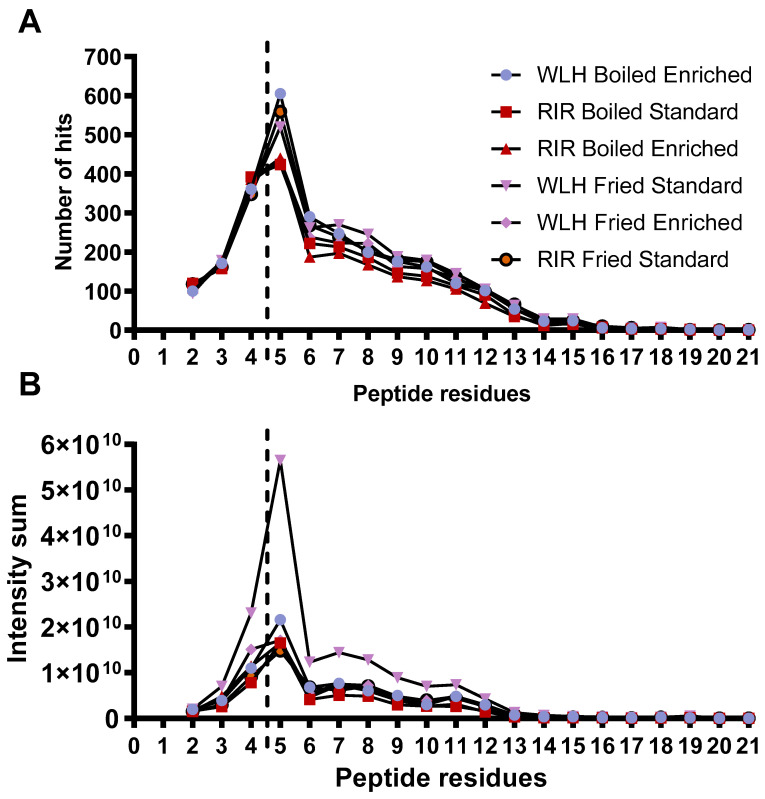
Peptide size distribution (**A**) by peptide residues measured through RPC and HILIC LC-MS/MS. Peptide size intensity (**B**) distribution was measured through RPC and HILIC LC-MS/MS. Note: Peptide length from two to four amino acid residues was analyzed through PEAKS studio de novo sequencing. Peptides with five and more residues were analyzed through PEAKS studio through the UniProt database. WLH: White Leghorn, RIR: Rhode Island Red.

**Figure 5 nutrients-13-04232-f005:**
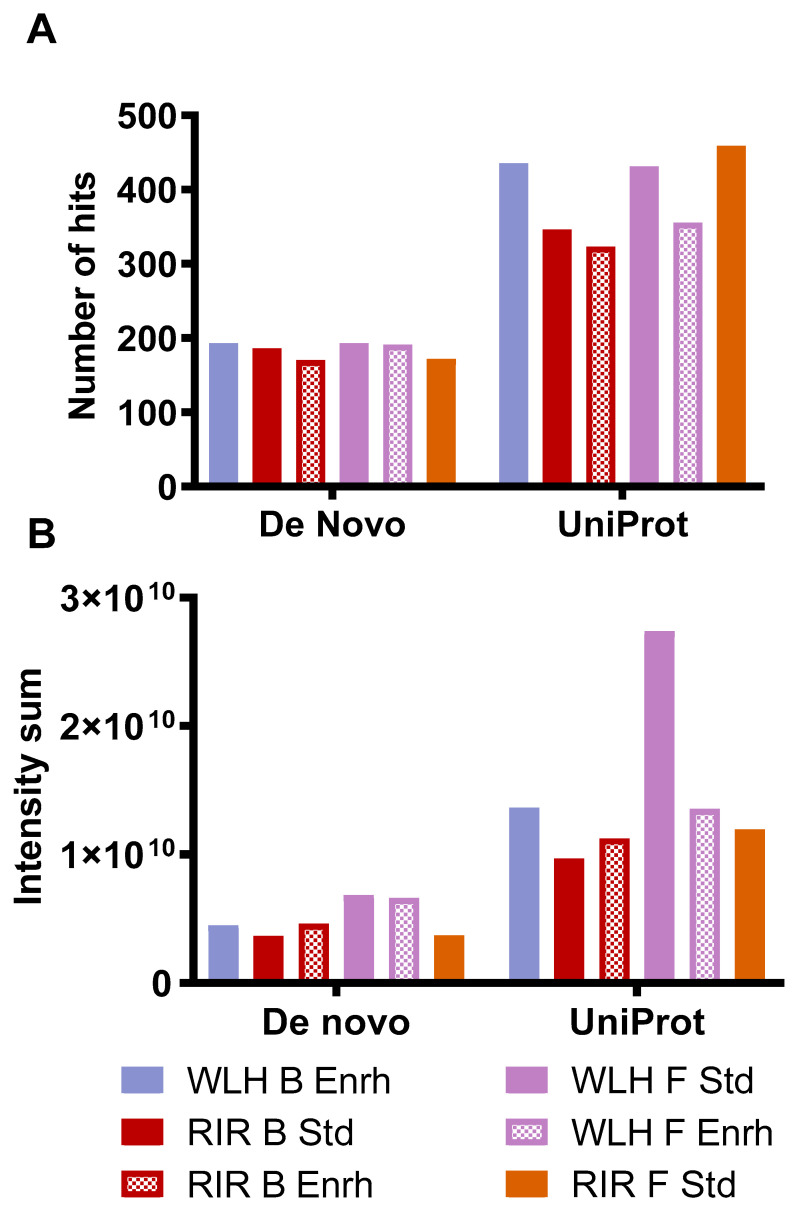
Number of hits (**A**) and intensity (**B**) of whole egg peptides with potential antioxidant activity-based structural requirements. Note: De novo includes peptides with two to four AA residues. Uniprot includes peptides with five or more AA residues. WLH: White Leghorn, RIR: Rhode Island Red, B: Boiled, F: Fried, Std: Standard, Enrh: Enriched.

**Table 1 nutrients-13-04232-t001:** Potential ACE Inhibitory peptides intensity sum present in the whole egg hydrolysates from whole egg and egg white proteins analyzed through RP and HILIC LC-MS/MS. Peptides complying with the ACE-inhibitory structural constraints (above the line) and previously reported ACE-inhibitory peptides (below the line) were observed in all the samples. MS2 spectra for each peptide is given in the Appendix A.

Peptide Sequence	WLH Boiled Enriched	RIR Boiled Standard	RIR Boiled Enriched	WLH Fried Standard	RIR Fried Standard	WLH Fried Enriched	Source	Enzyme Used	Reference
FF	5.68E+06	5.88E+06	-	9.78E+06	7.74E+06	7.91E+06	Whole egg	Pepsin-Pancreatin	-
YY	-	-	2.36E+06	4.59E+06	-	4.51E+06	Whole egg	Pepsin-Pancreatin	-
FY	-	4.53E+06	3.49E+06	-	3.16E+06	-	Whole egg	Pepsin-Pancreatin	-
VRFP	-	-	-	-	1.48E+06	-	Whole egg	Pepsin-Pancreatin	-
LW	2.01E+07	1.43E+07	-	1.19E+07	1.04E+07	1.30E+07	Ovalbumin	Pepsin	[44]
LY	-	-	6.41E+07	7.41E+07	-	-	Ovotransferrin	Chymotrypsin	[45]
NF	-	3.07E+06	5.09E+06	2.86E+06	3.25E+06	2.19E+06	Ovalbumin	Pepsin-Pancreatin	[46,47]
YR	2.54E+06	2.32E+06	1.13E+06	1.32E+06	1.64E+06	1.54E+06	Ovalbumin	Pepsin-Pancreatin	[46,48]
AW	-	1.46E+06	1.64E+06	-	1.34E+06	-	Lysozyme	Chymotrypsin-Thermolysin	[46]
MPF	4.69E+07	4.14E+07	9.30E+07	6.43E+07	5.12E+07	4.48E+07	Ovotransferrin	Pepsin-Pancreatin	[6]
ADHP	-	3.22E+06	7.62E+06	-	6.62E+06	6.75E+06	Ovalbumin	Pepsin-Pancreatin	[49]
Total	5.51E+07	6.19E+07	1.78E+08	1.57E+08	7.64E+07	6.77E+07			

Note: Di-, tri-, and tetrapeptides were analyzed through de novo sequencing. Peptides above the line complied with all the ACE-inhibitory structure constraints, and below the line complied in part with the structure constraints.

## Data Availability

Data available on request.

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
