# Peer review of "Bioactivity of Cooked Standard and Enriched Whole Eggs from White Leghorn and Rhode Island Red in Exhibiting In-Vitro Antioxidant and ACE-Inhibitory Effects"

_nutrients, 2021, doi:10.3390/nu13124232_

Round 1

Reviewer 1 Report

In the manuscript “Bioactivity of Cooked Standard and Enriched Whole Eggs from White Leghorn and Rhode Island Red in Exhibiting Antioxi dant and Antihypertensive Effectss”, the authors  evaluated the effect of hen breed, diet enrichment, and  simulated gastrointestinal digestion on the cooked whole egg-derived peptides in-vitro antioxidant and antihypertensive activities. The study is very interesting and the results are remarkable. The paper is well written and the experimental methods are adequate and fully support the research; there are only some issues that require clarification.

  • There are several popular assays being used to study anti-oxidative stress, such trolox equivalent antioxidant capacity assay (TEAC), total oxyradical scavenging capacity assay (TOSC), peroxyl radical scavenging capacity assay (PSC), and ferric reducing antioxidant potential assay (FRAP). Why the antioxidant capacity of whole egg hydrolysates was evalauted only through Oxygen Radical Absorbance Capacity (ORAC) assay.
  • Why are employed the Caco-2 cells for study the protection of intestinal digests against AAPH-induced oxidative stress and not the intestinal epithelial cells (IECs-6)?
  • Why used 2,2′-azobis(2-amidinopropane) dihydrochloride (AAPH) as source of free radicals and not hydrogen peroxide or LPS?
  • For measurement of in vitro antioxidant activity, the whole hydrolysate was ultrafiltrated to obtain the <3000 Da peptide fraction. Why the bioavailable peptides have not been purified from salts and sugars before cellular assays?
  • Eggs are source of radical scavengers such as phospholipids, carotenoids and Vitamin E. Why weren't these compounds monitored in the gastro intestinal digests?

Author Response

In the manuscript “Bioactivity of Cooked Standard and Enriched Whole Eggs from White Leghorn and Rhode Island Red in Exhibiting Antioxidant and Antihypertensive Effects”, the authors evaluated the effect of hen breed, diet enrichment, and simulated gastrointestinal digestion on the cooked whole egg-derived peptides in-vitro antioxidant and antihypertensive activities. The study is very interesting and the results are remarkable. The paper is well written and the experimental methods are adequate and fully support the research; there are only some issues that require clarification.

The authors would like to thank the reviewer for such encouraging comments. We have now made an attempt to address all the previous issues mentioned by the reviewer earlier.

There are several popular assays being used to study anti-oxidative stress, such Trolox equivalent antioxidant capacity assay (TEAC), total oxyradical scavenging capacity assay (TOSC), peroxyl radical scavenging capacity assay (PSC), and ferric reducing antioxidant potential assay (FRAP). Why the antioxidant capacity of whole egg hydrolysates was evaluated only through Oxygen Radical Absorbance Capacity (ORAC) assay.

Response: Although the antioxidant capacity of a bioactive compound can be measured in different ways through the previously mentioned assays, we only considered the ORAC antioxidant capacity method since it is the only one that evaluates the kinetics of the peroxyl radical quenching. It also incorporates the Trolox equivalent antioxidant capacity as the units of antioxidant capacity/g of the sample. We also used the ORAC assay keeping in mind the kinetic analysis through the in-vitro cell model. Thus to keep similarities between the assays we used ORAC.

Why are employed the Caco-2 cells for study the protection of intestinal digests against AAPH-induced oxidative stress and not the intestinal epithelial cells (IECs-6)?

Response: We thank the reviewer for bringing up this point as the type of cell line used for assays is often dependent to what’s available and the purpose of the study. The IECs-6 are cell lines derived from the brown rat’s intestinal crypt cells. In our study we wanted to use a physiological model and the Caco-2 are derived from human colorectal adenocarcinoma cells. Even though these cells are a disease model such as cancer, the physiological mechanism is similar to that one of healthy human gastrointestinal epithelial cells. Therefore, we proceed to use the Caco-2 cells. Finally, Caco-2 cells have been used successfully in previous studies evaluating the antioxidant activity of bioactive compounds such as phenols and peptides.

Why used 2,2′-azobis(2-amidinopropane) dihydrochloride (AAPH) as source of free radicals and not hydrogen peroxide or LPS?

Response: The question refers to an important aspect of the antioxidant protocol. Hydrogen peroxide and 2,2′-azobis(2-amidinopropane) dihydrochloride (AAPH) have been previously used in the cellular antioxidant activity assay. We used AAPH since it is a stronger inducer of peroxyl radicals compared to hydrogen peroxide and LPS. The two latter ones are more physiologically relevant, but it is hard to detect differences as the signal obtained from them is not as high as AAPH. Using LPS is a good suggestion for further improvement of the model, but it is dependent on the MyD88 pathway for the production of cytokines which induces oxidative stress could be a potential concern. The cascade could be different in an in-vitro cell model and needs to be validated in future experiments.

For measurement of in vitro antioxidant activity, the whole hydrolysate was ultrafiltrated to obtain the <3000 Da peptide fraction. Why the bioavailable peptides have not been purified from salts and sugars before cellular assays?

Response: The results in our previous manuscript (DOI: 10.1016/j.foodchem.2020.128623) suggest that peptides constitute a 30-40% of the whole hydrolysate. We do not purify the ultrafiltrate fraction as we want to model the physiological conditions in the small intestine which can also absorb the sugars and salts present.

Eggs are source of radical scavengers such as phospholipids, carotenoids and Vitamin E. Why weren't these compounds monitored in the gastrointestinal digests?

Response: It is well known that egg is a source of the mentioned radical scavengers. In our previous study, we evaluated the content of xanthophylls (antioxidant carotenoids) in the cooked egg and the whole hydrolysate (DOI: 10.1016/j.foodchem.2020.128623). Phospholipids and vitamin E were not monitored as they were not compounds directly enriched from the hen’s diet in contrast to the omega-3 and carotenoids which were enriched through marigold and flaxseed present in the diet. We will evaluate in the future the phospholipids and vitamin E content in the whole hydrolysate which can influence the ORAC antioxidant capacity in further studies. But the primary focus for this study was peptides.

Reviewer 2 Report

The manuscript reports the bioactivity (antioxidant and antihypertensive) of eggs from different hen breeds, cooking modes and subjected to in vitro simulated gastrointestinal digestion.

Despite the manuscript being well written, I find the topic is not scientifically sound. Besides, there is not very innovative in any of the methods reported by the authors, and the graphs and presentation of data are not good.

Author Response

Despite the manuscript being well written, I find the topic is not scientifically sound. Besides, there is not very innovative in any of the methods reported by the authors, and the graphs and presentation of data are not good.

Response: We thank the reviewer for positive feedback on the writing quality. We would appreciate it if the reviewer can provide additional feedback and be more specific on the reasons why the topic is not scientifically sound. There have been several studies evaluating the antioxidant and ACE-inhibitory/anti-hypertensive activities of bioactive molecules once released from their parent protein while being evaluated through in-vitro and in vivo models as observed in references 6, 8, 11, 13, 15, 18, 19, 26, 35, 36, 40, 42, 43, 47, 52. Even though the methods are based on previous research (which are already validated), the innovation of the study relies on evaluating the combination of the independent factors and evaluating if there is any interaction between hen breed, cooking methods, and diet. Previous research focused only on the effect of one or two factors. Also, the manuscript is a follow-up study of a previously published article that was reviewed and accepted by the scientific community (Reference: DOI: 10.1016/j.foodchem.2020.128623). Also, we want to mention that through a literature search we did not come across a single study with the whole egg that evaluated the bioactivity considering all these factors- hen breed, cooking methods, and diet. We hope future reviews will enable us to improve our research and manuscripts following constructive criticism.

Reviewer 3 Report

Dear Authors,

The title of Your manuscript is misleading. The study is in vitro and therefore You did not show any antihypertensive results. Please consider changing the title.

Chapter 2.2 sample preparation:

There is just a short information forwarding to Your last paper. To be fair there should be at least a short depiction of the changes in the feeding of the chickens and parameters of the cooking methods.

There is no description of the GI digestion method - pleas consider a short description.

Chapter 2.5 ACE inhibition:

You presented a value based on information from a last paper. This can be misleading when reading : ug of peptide/mL. That is suggesting that You measured the amount of a particular peptide. Pleas consider inserting the results of the whole hydrolysate concentrations and peptide content.

Chapter 2.6 Peptide identification

The fragmentation method was not described (DDA, DIA...?). Pleas show a MS spectrum with identified fragment ions.

In line 214 You stated that the ALC cutoff was 50%. Why did You choose such a low parameter? Pleas insert a table with identified peptides in PEAKS Studio.

Chapter 3.3

Line 275 - the unit "ug of peptide/mL" is related to what? Was the analysis carried out with the whole egg hydrolysate or a peptide extract?

Line 277 - the sentence is in my opinion incomplete "[...] hen breed." Maybe: [...] hen breed and cooking method.

Line 280-282 - how is this sentence related to Your findings?

Chapter 3.4:

Line 308-310. The sentence is not finished.

Page 11, table 1:

Pleas consider filling the blank places (not identified) with a sign (i.e. "-") for better understanding. Furthermore the line in the table is confusing. Are the peptides above novel/unique? 

Why did You tetrapeptides analyze wit Uniprot and not with Peaks?

Chapter 4:

Line 394-396 the sentence is unclear.

Line 425 You mentioned a peptide profile that was not shown in the manuscript. Maybe a supplement with this data would be helpful.

Line 443-445 You clearly can not state that any activity is due to the presence of any one molecule, because You analyzed a whole egg hydrolysate. Please rephrase this sentence.

Concluding

The manuscript is well prepared and written. There are some issues with data presentation and a lack of certain data, but overall the since behind the data seems sound. My recommendation is to publish after minor changes and the addition of missing data.

Author Response

The title of Your manuscript is misleading. The study is in vitro and therefore You did not show any antihypertensive results. Please consider changing the title.

Response: The title has been modified to implement the reviewer’s suggestion.

‘Bioactivity of Cooked Standard and Enriched Whole Eggs from White Leghorn and Rhode Island Red in Exhibiting In Vitro Antioxidant and ACE-inhibitory Effects’

Chapter 2.2 sample preparation:

There is just a short information forwarding to Your last paper. To be fair there should be at least a short depiction of the changes in the feeding of the chickens and parameters of the cooking methods.

There is no description of the GI digestion method - please consider a short description.

Response: The details of the diet and cooking methods were incorporated in section 2.2 as detailed in the lines below:           

‘L93-106. The hens' diet consisted of a corn-soybean-based diet without (standard) and with flaxseed and ORO GLO® as a source of omega-3, lutein, and zeaxanthin (enriched). The hens were fed weekly for 16 weeks. Next, whole eggs were boiled or fried and subjected to simulated GI digestion following the Minekus et al. (2014) and Mat et al. (2016) methods with modifications according to our previous studies [10,24,25]. Eggs were boiled in a saucepan with water covering up to 2.5 cm above the eggshell. After 10 min the eggs were removed and cooled off in running tap water for five minutes. The egg was peeled for storage at -20 °C under vacuum in a vacuum bag. Fried eggs were previously homogenized and cooked in a pan fryer heated at 191 °C (350 °F) for 90 s each side. The eggs were cooled for 15 min and stored at -20 °C under vacuum in a vacuum bag. The simulated digestion consisted of an oral phase using α-amylase and simulated salivary fluids at pH 7 for 2 min, a gastric phase using pepsin and simulated gastric fluid at pH 3 for 2 h, and intestinal digestion using pancreatin and simulated intestinal fluid at pH 7 for 2 h.’

Chapter 2.5 ACE inhibition:

You presented a value based on information from the last paper. This can be misleading when reading : ug of peptide/mL. That is suggesting that You measured the amount of a particular peptide. Please consider inserting the results of the whole hydrolysate concentrations and peptide content.

Response: We thank the reviewer for suggesting to present the data as a whole hydrolysate/mL concentration. The ACE-inhibition results has been modified to show the IC50 based on the whole hydrolysate and not on the peptide content. A revised figure has been submitted and appended to the revised document (filename: Paper 3_MS_Antioxidant whole egg_Figure3_v2). The change had an effect on the statistical results as RIR fried standard ACE-inhibition is significantly different to all the treatments. However, the change does not influence the overall results and discussion. Please see the modified version of section 2.5 L175-177 and section 3.3, L279-295.

The figure caption in L301 was modified to correct the units as µg of the whole hydrolysate/mL.

‘L301: Figure 3. Whole egg hydrolysates ACE Inhibition IC50 (µg of whole hydrolysate/mL); Error bars represent standard error’.

Chapter 2.6 Peptide identification

The fragmentation method was not described (DDA, DIA...?).

Response: Data acquisition mode and method of fragmentation are now given in lines 199-203 (please see amended text below).

‘Line 190-193. The data was acquired on a Q Exactive™ HF mass spectrometer (MS) (Thermo Fisher, Waltham, MA, USA) operating in data-dependent acquisition (DDA) mode, using a mass range of 60 to 750 m/z on single charged ions. The isolated ions were further fragmented using collision-induced dissociation (CID) of an isolation window of 2 m/z.’

Please show a MS spectrum with identified fragment ions.

Response: The MS2 spectra for each peptide listed in Table 1 has been added in Supplementary Data 1 (S. Figure 2). Quality criteria and other information for all peptides identified are given in Supp data 2. The information is disclosed in section 3.4:

‘L358-L361: The MS2 fragmentation profile of the identified peptides is included in S. Figure 2. A minimum ALC of 50% and a parent ion mass accuracy of less than ±5 ppm was obtained for all the peptides.’

Chapter 3.3

Line 275 - the unit "ug of peptide/mL" is related to what? Was the analysis carried out with the whole egg hydrolysate or a peptide extract?

Response: Following the comment of section 2.5 the units have been modified to µg of whole hydrolysate/mL.

‘L282-284: The sample with the highest ACE inhibitory activity (lowest IC50) value was observed in RIR boiled enriched egg (IC50: 221 µg of whole hydrolysate/mL) (Figure 3).’

‘L286-287: Regarding WLH, fried enriched eggs showed the highest ACE inhibition activity value (IC50: 272 µg of whole hydrolysate/mL) (Figure 3).’

Line 277 - the sentence is in my opinion incomplete "[...] hen breed." Maybe: [...] hen breed and cooking method.

Response: Modified as per suggestion:

‘L284-286: The RIR boiled enriched egg high ACE-inhibitory activity and antioxidant activity compared to RIR fried enriched egg suggest a possible synergistic activity of the hen breed and cooking method.’

Line 280-282 - how is this sentence related to Your findings?

Response: We included a new reference, Majumder and Wu, 2009 to support our previous statement that ‘these samples are expected to have shorter peptides, a characteristic often found in ACE-inhibitory peptides’. The study on white shell standard fried eggs, GI-digested found that it was a source of short-length (low molecular weight) anti-hypertensive peptides. These peptides may also be present in our samples since they are from the same source.

‘L288-291: These samples are expected to have shorter peptides, a characteristic often found in ACE-inhibitory peptides [31]. For example, white shell standard fried eggs subjected to GI digestion in a 5% (w/v) slurry heated at 80 °C showed the presence of tri- and pentapeptides with antihypertensive activity in fried whole egg digest [6]’.

Chapter 3.4:

Line 308-310. The sentence is not finished.

Response: We have modified the sentence to show examples of peptides with antioxidant activity once release from its parent protein, in this case, ovalbumin. Please see the correction in the modified word version L324-326.

‘L316-318: Several studies have reported the antioxidant activity of peptides once released from ovalbumin such as AEERYP, DEDTQAMP, DSTRTQ, DKLPG, DVYSF, and ESKPV [16,42].’

Page 11, table 1:

Please consider filling the blank places (not identified) with a sign (i.e. "-") for better understanding. Furthermore, the line in the table is confusing. Are the peptides above novel/unique?

Response: The table has been modified as per suggestion. The peptides above the line complied with all the ACE-inhibitory structure constraints and below the line complied in part with the structural constraints. The information is detailed in the table header and footnote for the reader.

Why did You tetrapeptides analyze wit Uniprot and not with Peaks?

Response: The footnote of the peptide has been corrected to indicate that di, tri, and tetrapeptides were all analyzed using de novo sequencing.

Chapter 4:

Line 394-396 the sentence is unclear.

Response: The line has been modified to indicate that high molecular weight peptides can potentially exhibit antioxidant properties in a chemical method rather than in a cell-based model since no bioavailability is taken into account in a chemical method.

‘L408-410: Rao et al. 2020 suggested that high molecular weight peptides from egg white can potentially exhibit a higher antioxidant activity in ORAC due to their radical quenching capacity independently of their bioavailability[35].’

Line 425 You mentioned a peptide profile that was not shown in the manuscript. Maybe a supplement with this data would be helpful.

Response: The sentence was modified to indicate a peptide size distribution, not profile. The description of each peptide identified in the hydrolysate is given in Supplementary Data 1.

‘L439-441: The peptide size distribution showed no difference of small MW peptides number and intensity through de novo analysis complying with the antioxidant structural requirements (Figure 4A and 4B).’

Line 443-445 You clearly can not state that any activity is due to the presence of any one molecule, because You analyzed a whole egg hydrolysate. Please rephrase this sentence.

Response: The sentence has been rephrased.

‘L442-444: It remains to be elucidated if the ACE-inhibitory capacity is derived from the individual peptides or by a synergistic effect of the peptide’s mixtures along with other molecules present in the whole hydrolysate.’